# Long-Term Impact of COVID-19 on Mental Health among the General Public: A Nationwide Longitudinal Study in China

**DOI:** 10.3390/ijerph18168790

**Published:** 2021-08-20

**Authors:** Le Shi, Zheng-An Lu, Jian-Yu Que, Xiao-Lin Huang, Qing-Dong Lu, Lin Liu, Yong-Bo Zheng, Wei-Jian Liu, Mao-Sheng Ran, Kai Yuan, Wei Yan, Yan-Kun Sun, Si-Wei Sun, Jie Shi, Thomas Kosten, Yan-Ping Bao, Lin Lu

**Affiliations:** 1Peking University Sixth Hospital, Peking University Institute of Mental Health, NHC Key Laboratory of Mental Health (Peking University), National Clinical Research Center for Mental Disorders (Peking University Sixth Hospital), Peking University, Beijing 100191, China; leshi@bjmu.edu.cn (L.S.); 1911210616@bjmu.edu.cn (Z.-A.L.); quejianyu@bjmu.edu.cn (J.-Y.Q.); weijian191954@stu.pku.edu.cn (W.-J.L.); yuankai@pku.edu.cn (K.Y.); weiyan@bjmu.edu.cn (W.Y.); yankunsun@bjmu.edu.cn (Y.-K.S.); siwei341105@bjmu.edu.cn (S.-W.S.); 2Savaid Medical School, University of Chinese Academy of Sciences, Beijing 100049, China; huangxiaolin18@mails.ucas.ac.cn; 3Beijing Key Laboratory of Drug Dependence, National Institute on Drug Dependence, Peking University, Beijing 100191, China; 1410306211@pku.edu.cn (Q.-D.L.); liuln@pku.edu.cn (L.L.); shijie@bjmu.edu.cn (J.S.); 4School of Public Health, Peking University, Beijing 100191, China; 5Peking-Tsinghua Center for Life Sciences, PKU-IDG/McGovern Institute for Brain Research, Beijing 100871, China; yongbozheng@bjmu.edu.cn; 6Department of Social Work and Social Administration, University of Hong Kong, Hong Kong, China; msran@hku.hk; 7Department of Psychiatry, Baylor College of Medicine, Houston, TX 77030, USA; kosten@bcm.edu; 8Research Unit of Diagnosis and Treatment of Mood Cognitive Disorder (2018RU006), Chinese Academy of Medical Sciences, Beijing 100191, China

**Keywords:** COVID-19, mental health, longitudinal, China

## Abstract

COVID-19 might have long-term mental health impacts. We aim to investigate the longitudinal changes in mental problems from initial COVID-19 peak to its aftermath among general public in China. Depression, anxiety and insomnia were assessed among a large-sample nationwide cohort of 10,492 adults during the initial COVID-19 peak (28 February 2020 to 11 March 2020) and its aftermath (8 July 2020 to 8 August 2020) using the Patient Health Questionnaire-9, Generalized Anxiety Disorder-7, and Insomnia Severity Index. We used generalized estimating equations and linear mixed models to explore factors associated with long-term mental health symptoms during COVID-19. During the five months, mental health symptoms remained consistently elevated (baseline 46.4%; follow-up 45.1%). Long-term depression, anxiety and insomnia were associated with several personal and work-related factors including quarantine (adjusted OR for any mental health symptoms 1.31, 95%CI 1.22–1.41, *p* < 0.001), increases in work burden after resuming work (1.77, 1.65–1.90, *p* < 0.001), occupational exposure risk to COVID-19 (1.26, 1.14–1.40, *p* < 0.001) and living in places severely affected by initial COVID-19 peak (1.21, 1.04–1.41, *p* = 0.01) or by a COVID-19 resurgence (1.38, 1.26–1.50, *p* < 0.001). Compliance with self-protection measures, such as wearing face masks (0.74, 0.61–0.90, *p* = 0.003), was associated with lower long-term risk of mental problems. The findings reveal a pronounced and prolonged mental health burden from the initial COVID-19 peak through to its aftermath in China. We should regularly monitor the mental health status of vulnerable populations throughout COVID-19.

## 1. Introduction

Despite the incidence of over 100 million COVID-19 infections and three million deaths from around the world, the pandemic is far from being controlled [1]. Because of the high transmission rate and severe pathology from the newly emerged coronavirus, infected cases and deaths continue to soar despite widespread public health measures, including lockdowns, quarantine, social distancing and business closures [2]. The pandemic is expected to last for an extended period of time, as long-term effectiveness of vaccines and other control methods are yet to be determined [3]. Thus, the dynamic impact of COVID-19 should be deeply researched.

COVID-19 and its corresponding economic and social burden have a negative immediate impact on mental health [4,5,6]. The prevalence of depression, anxiety and insomnia during the initial COVID-19 outbreak was approximately 30% both in China and abroad, which was substantially higher compared with 6.0% for depression, 5.3% for anxiety and 15.0% for insomnia before the COVID-19 outbreak [4,7,8,9,10,11,12,13]. Psychological risk is disproportionately distributed across demographic subgroups during initial COVID-19 peak. For instance, men show higher risk for mental health problems than women, and public health measures, including quarantine, lockdowns and social distancing appear to increase risk for mental health problems [9,14,15,16,17]. Moreover, previous studies after SARS indicated prolonged distress after exposure to the pandemic, and its long-term influencing factors include demographics (i.e., gender and age), personal factors associated with SARS (i.e., SARS infection, quarantine and social distancing), and work-related factors (i.e., direct engagement in SARS control and occupational exposure to SARS) [18,19,20,21]. Existing longitudinal studies have identified a persistent mental health burden under the substantial and enduring strain of COVID-19 in many countries, including the United States, the United Kingdom, and Italy, but little evidence exists in China [22,23,24,25]. In China, the main wave of COVID-19 was largely put under control and lockdown was lifted in April, 2020. However, sporadic cases and local resurgences have occurred continuously across the country, and nationwide epidemic prevention and control have been continuously enforced to suppress another large-scale outbreak. Thus, continuous emergence of sporadic cases, persistence of public health measures, sustained economic stress and an uncertain efficacy of potential treatment agents still pose considerable threats to mental health [26,27,28].

Longitudinally tracking mental health changes is essential for mental health management [29], but most studies on the psychological impact from COVID-19 are cross-sectional or conducted within the initial peak or lockdown period and thereby fail to capture changing trends across different pandemic stages of COVID-19. Additionally, though many factors are associated with immediate mental health symptoms, these factors can induce sustained mental health changes, and new risks or protective factors may emerge after the initial peak, when the lockdown is lifted, and economic and social activity restart. Given the continuous emergence of sporadic cases, sustained pandemic control measures and remarkable social disruptions due to COVID-19, we hypothesize that severe mental health symptoms persist from initial COVID-19 peak to its aftermath, and factors for long-term distress may differ from those for immediate stress responses to COVID-19. Therefore, the objective of this research is to explore longitudinal changes in mental health symptoms and their associated factors from initial COVID-19 to its aftermath, by repeatedly measuring mental health symptoms among a national cohort in China.

## 2. Methods

### 2.1. Study Design

We conducted a longitudinal cohort study, in which participants were recruited from the health page of Chinese website Joybuy. Joybuy is a large ecommerce and information service platform that provides online health products and services in China with 0.44 billion active users from all 34 province-level regions in China, the members of which are generally young and highly educated, who acquire membership by online registration with an annual fee [9,17]. During the baseline survey fielded from 28 February 2020 to 11 March 2020, which was the initial peak of COVID-19, all registered members were allowed to click on a link on health page of website Joybuy to complete the survey until the convenience sample covered all 34 province-level regions in China. The follow-up survey was fielded from 8 July 2020 to 8 August 2020, when the main COVID-19 wave had been basically controlled, and lockdown had been lifted for about three months in China. At the follow-up survey, we adopted two methods to recruit participants. Firstly, questionnaire links were sent via the message platform of Joybuy to all baseline participants, who were identified by their unique ID numbers. At the same time, to recruit new participants, we put our second-wave survey link on the health page of Joybuy, allowing new participants to voluntarily click on it to participate the second-wave survey. At both the baseline and follow-up survey, links were attached with online shopping vouchers. To allow for comparison between baseline and follow-up, participants who responded to both surveys were included in the final analysis in this study. The study was approved by the ethics committee of Peking University Sixth Hospital (Institute of Mental Health) (ethical code: 2020-2-21-2). Informed consent was received online before the respondents began the surveys. This study follows the American Association for Public Opinion Research (AAPOR) reporting guideline.

### 2.2. Participants

The participants in the two surveys were all registered members of Joybuy. At baseline, 56,679 adults providing valid age information were included, as detailed elsewhere [9]. At the follow-up, of the 56,679 baseline participants to whom we sent follow-up survey links, 17,576 clicked on the links and 10,867 commenced the survey. Finally, 10,492 participants from 32 provinces in China provided informed consent and completed the follow-up survey (effective follow-up rate: 18.5%). Data from the two surveys were matched individually according to unique ID numbers from the Joybuy website. Detailed information about recruitment and selection of participants in the follow-up survey are presented in Figure 1.

### 2.3. Measures and Variables

Each survey lasted ~20 min and had four parts. The first part gathered demographic information of the participants. The second part asked epidemic-related questions. The third part evaluated quarantine conditions and social attitudes toward the COVID-19 pandemic. We provided detailed information about these three parts previously [9]. The fourth part consisted of three standardized scales, including the Chinese versions of Patient Health Questionnaire-9 (PHQ-9), Generalized Anxiety Disorder-7 (GAD-7), and Insomnia Severity Index (ISI), that measured symptoms of depression, anxiety, and insomnia, respectively. PHQ-9 is a 9-item questionnaire based on the diagnosis of a major depressive episode according to the DSM systems with a Cronbach’s α of 0.89, sensitivity of 0.88 and specificity of 0.88 to detect depression [30]. GAD-7 is a 7-item questionnaire considered as a reliable tool to detect anxiety disorder with a Cronbach’s α of 0.89, sensitivity of 0.89 and specificity of 0.82 [31]. ISI is a 7 item questionnaire with a Cronbach’s α of 0.92, and is found to produce a sensitivity of 0.82 and a specificity of 0.82 in detecting clinical insomnia [32]. We calculated participants’ scores and used cutoff scores of 5, 5, and 8 to categorize them as depressed, anxious and having insomnia symptoms. We added extra questions specific to the follow-up survey to inquire about new situations after the initial peak of COVID-19, including COVID-19 resurgences, whether wearing face masks voluntarily when going out, whether reducing gatherings voluntarily, whether seeking psychological consultation, and self-perceived increases in work burden after resuming work. The summary of the questionnaire questions for both surveys is provided in Appendix A.

We considered three groups of potential factors associated with long-term mental health symptoms: (1) Demographics: gender, age, living area, educational level, marital status, monthly family income, history of chronic diseases, history of psychiatric disorders, and family history of psychiatric disorders. (2) Personal factors associated with COVID-19: personal infection, family members infection, living in the province most severely affected by initial peak, experiencing quarantine, living in places with a COVID-19 resurgence, wearing face masks voluntarily when going out, reducing gatherings voluntarily and seeking psychological consultation. (3) Work-related factors: direct engagement in work-related COVID-19 control, self-perceived occupational exposure risk to COVID-19, and self-perceived increase in work burden after resuming work. Appendix A presents detailed descriptions for variable constructing approaches based on participants’ responses to questionnaire questions on these factors.

### 2.4. Statistical Analyses

Descriptive statistics were used to present the baseline demographic characteristics of the total baseline sample, participants who responded to the follow-up survey and those who did not respond. Weight based on gender (male or female), age (18–39 years or ≥40 years), living area (urban or rural), educational level (lower than college school or college school or higher), marital status (married or unmarried), geographical regions and history of chronic diseases (yes or unknown/no) in the baseline sample was applied to adjust for proportion differences between the baseline total sample and the longitudinal cohort. Prevalence of symptoms of depression, anxiety, insomnia, and any mental health problems at both surveys was calculated using the aforementioned cutoff scores, and reported as the percentages of cases. 95% confidence intervals (CIs) were produced by the exact binomial methods. Average scores of PHQ-9, GAD-7 and ISI at both surveys were presented as medians and interquartile ranges (IQRs). McNemar *χ^2^* tests and paired-samples Wilcoxon tests were respectively adopted to test the statistically significant differences in prevalence and scores of mental health symptoms between baseline and follow-up survey. Proportions of new onset and persistent mental health symptoms were calculated among participants who were screened as symptom negative or positive at baseline, respectively.

To explore the associated factors of long-term mental health symptoms, we performed two analyses. Generalized estimating equation models with a binomial distribution were employed, in which the outcome variables were the categorical mental health status (yes/no), with survey order as repeated effect and within-subject effect, and participant ID number as covariate factor. All potential factors were first entered into a univariable model to test for significance, and then a multivariable model was constructed by including all potential factors that showed significance in their individual models, as well as the variable of survey order. We chose the autoregressive correlation structure based on lower values for the Quasi Information Criterion. In addition, we fitted generalized linear mixed models with random within-subject intercepts and autoregressive covariance structures, in which the outcome variables were PHQ-9, GAD-7 and ISI scores. Fixed effects for all potential factors considered were first tested for significance in their individual models. We also tested the fixed effects for all factors with time interactions to explore factors associated with longitudinal mental health changes from baseline to follow-up. The multivariable model was constructed by including fixed effects for all potential factors and factor by time interactions that showed significance in individual models, with time coded as 0 at baseline and 5 at follow-up. We also tested models with both the random slopes and intercepts, and other covariance structures, and selected the final optimal model based on lower Akaike information criterion and likelihood ratio tests.

The level of significance was set to two sided *p* < 0.05. All of the statistical analyses were performed using SPSS 22 software (SPSS, Chicago, IL, USA) and R version 4.0.3.

## 3. Results

### 3.1. Demographic and Epidemic-Related Characteristics of Participants

Among the 10,492 participants included in the longitudinal cohort, the mean (SD) age was 36.87 (8.21), and 4465 (42.6%) were male, 9769 (93.1%) lived in urban areas, 8391 (80.0%) had a college school or higher educational level, 8467 (80.7%) were married, and 2444 (23.3%) had family monthly income lower than 5000 yuan. Compared to those who did not participate in the follow-up survey, those who took part in the follow-up survey were more likely to be female, older, poorly educated, married and report history of chronic diseases. Detailed baseline characteristics of the longitudinal cohort, the total baseline sample and weighted longitudinal cohort sample are shown in Table 1.

### 3.2. Change in Mental Health Symptoms from Baseline to Follow-Up Survey

The weighted prevalence of any mental health symptoms was continuously elevated from the baseline survey at 46.4% (95%CI 45.4–47.3%) to the follow-up survey at 45.1% (44.2–46.1%), although a slight but statistically significant decline was detected (*p* = 0.01). Depression significantly increased from 30.0% (29.2–30.9%) to 33.6% (32.7–34.5%, *p* < 0.001), and insomnia significantly increased from 29.8% (28.9–30.7%) to 35.3% (34.4–36.2%, *p* < 0.001). In contrast, anxiety symptoms decreased from 35.2% (34.3–36.1%) to 32.5% (31.7–33.5%, *p* < 0.001). Medians (IQR) of PHQ-9 scores increased from 0.0 (0.0–6.0) to 0.0 (0.0–8.0, *p* < 0.001), and ISI scores increased from 4.0 (1.0–8.0) to 4.0 (1.0–9.0, *p* < 0.001). In contrast, GAD-7 scores decreased from 1.0 (0.0–7.0) to 0.0 (0.0–7.0, *p* < 0.001). Table 2 shows the changes in prevalence and scores of mental health symptoms in the cohort, and changes in scores of depression, anxiety and insomnia from baseline to follow-up are shown in Appendix A.

We present mental health prevalence changes stratified by demographic, personal and work-related factors in Figure 2 and Appendix A. Mental health symptoms declined from baseline to follow-up among females, mid-aged or elder people, urban residents, those poorly educated, participants with moderate or high incomes and married participants, while we identified no statistically significant difference in prevalence of any mental health symptoms from baseline to follow-up among males, young adults, rural residents, the highly educated, and impoverished participants.

Mental health symptoms also declined between baseline and follow-up in participants who were not family members of COVID-19 participants, lived in places most severely affected by initial peak, did not have quarantine experiences, occupational exposure risk, COVID-19 resurgence experiences or increases in work burden, and in participants who wore face masks, reduced social gatherings voluntarily or did not seek psychological consultation during follow-up. However, sustained mental health symptoms were observed among family members of COVID-19 patients, participants with quarantine experiences, with occupational exposure risk to COVID-19, with increases in work burden after resuming work, and those living in places with COVID-19 resurgences, and those who did not wear face masks when going out or did not reduce social gatherings. Within these subgroups, the prevalence of any mental health symptom at both surveys remained over 40%, with no statistically significant difference observed between baseline and follow-up survey (all *p* > 0.05).

### 3.3. Proportions of New Onset and Persistent Mental Health Symptoms from Baseline to Follow-Up Survey

At baseline 5627 participants reported no mental health symptom, and at follow-up 1374 (24.4%) displayed new onset symptoms. At baseline 4865 participants reported any mental health symptoms, and at follow-up 3359 (69.0%) reported persistent symptoms. Specifically, of participants denying depression, anxiety or insomnia at baseline, 20.1%, 18.5% and 21.2% displayed new onset depression, anxiety or insomnia symptoms at follow-up. Among the participants reporting depression, anxiety or insomnia at baseline, 65.1%, 58.5% and 68.4% reported persistent depression, anxiety and insomnia symptoms. The proportions of participants with new onset and persistent mental health symptoms are shown in Table 3.

### 3.4. Factors Associated with Long-Term Positive of Mental Health Symptoms during the Course of COVID-19

In the multivariable generalized estimation equation analysis, some personal factors were associated with all four of the long-term mental health symptoms and are presented in Table 4, including living in places most severely affected by the initial peak (adjusted OR for any mental health symptoms 1.21, 95%CI 1.04–1.41, *p* = 0.01) or with COVID-19 resurgences (1.38, 1.26–1.50, *p* < 0.001), and quarantine experience (1.31, 1.22–1.41, *p* < 0.001). Moreover, some work-related factors also showed significance in the analysis, including occupational exposure risk to COVID-19 (1.26, 1.14–1.40, *p* < 0.001), and increases in work burden after resuming work (1.77, 1.65–1.90, *p* < 0.001). Family members of COVID-19 patients (2.10, 1.61–2.74, *p* < 0.001) and people seeking psychological consultation since COVID-19 (2.57, 2.33–2.85, *p* < 0.001) also emerged as susceptible populations. Wearing facemasks voluntarily (0.74, 0.61–0.90, *p* = 0.003) appeared to protect from long-term mental health symptoms. Factors associated with long-term depression, anxiety, insomnia and any mental health symptoms are presented in Table 4.

### 3.5. Factors Associated with Scores of Depression, Anxiety and Insomnia Symptoms during the Course of COVID-19

Table 5 presents the multivariable generalized linear mixed-effects analysis with outcome variables being continuous PHQ-9, GAD-7 and ISI scores, and the fixed effects for all factors detected significant in the multivariable generalized estimating equation analysis remained statistically significant. Furthermore, we found that the highly educated, those living in places with COVID-19 resurgences, those reporting increases in work burden after resuming work and those seeking psychological consultation since COVID-19 experienced a steeper increase in depressive, anxiety or insomnia symptoms, while those who wore facemasks or reduced social gatherings voluntarily experienced a milder increase in the three symptoms over the course of COVID-19 (Appendix A), as indicated by significant factor by time interactions in the multivariable generalized linear mixed-effects models.

## 4. Discussion

This large-sample national cohort in China investigated the longitudinal changes in mental health symptoms from the initial peak of COVID-19 to its aftermath five months later. In both baseline and follow-up surveys, over 40% participants reported mental health symptoms, suggesting a consistently severe mental health burden during the COVID-19 pandemic. Our study further found about a quarter new onset and seven over ten persistent mental health symptoms among those without or with mental health symptoms at baseline. We identified personal (quarantine and living in places severely affected by initial peak or hit by COVID-19 resurgences) and work factors (occupational exposure risk to COVID-19 and increases in work burden after resuming work) to be associated with long-term mental health symptoms. In addition, special attention should be paid to family members of COVID-19 patients and those seeking psychological consultation. Some self-protection measures (wearing face masks and reducing social gatherings) were identified as potential protective factors for mental health symptoms. Given the continuous emergence of new cases and normalization of pandemic control, these findings can be valuable for long-term mental health management in the worldwide and persistent battle with COVID-19.

In other countries, longitudinal studies reported inconsistent temporal trends in mental health symptoms during COVID-19, but we found consistently elevated mental health symptoms in China. The UK reported a pronounced increase in distress immediately after the COVID-19 outbreak, followed by a rapid recovery [23,33]. Italy demonstrated a worsening trend in mental health during lockdown, while during the initial peak in the United States, psychological distress remained largely stable [24,25]. All these studies just focused on the changes in mental health status within the initial peak and lockdown period, while our study tracked the mental health changes from the initial peak to its aftermath, when the initial outbreak had been put under control, and lockdown had been lifted for about three months. Specifically, in this cohort study, 69.0% of participants with distress at baseline still reported distress at follow-up, while new cases of mental health symptoms emerged in 24.4% of the baseline symptom-free participants. The prolonged deterioration in mental health, despite an improvement in pandemic control, might be due to prominent social repercussions including pervasive public health measures and economical stress, as well as the frequent COVID-19 resurgences [26,27,28].

Within all demographic subgroups, the prevalence of mental health symptoms remained largely unchanged over time, although this symptom persistence was more evident among males and young people, as well as highly educated, impoverished and unmarried populations. Young adults’ long-term distress may be attributable to their higher exposure to social media and misinformation, more significant increases in work burden as well as more insecurity in jobs and finance [34,35]. Well-educated people harbor more interest in health-related information, thus are more likely to be psychologically influenced by COVID-19 [36]. Impoverished people may be less capable of coping with the financial adversities caused by the economic downturn following COVID-19, which can breed long-term distress [33].

Several personal and work-related factors were associated with long-term mental health symptoms. Personal factors mainly included quarantine and living in places severely affected by initial peak or hit by COVID-19 resurgences. Quarantine has become a regular international intervention for COVID-19 [37], but quarantine induces lonely feelings and financial damage, with long-term mental health consequences [38,39]. Shorter quarantine duration, simplification of quarantine procedures and more knowledge about the benefits of quarantine can lower long-term psychological risk [40]. Apart from quarantine, this study combined with our previous finding [9] suggests that people living in the place most severely affected by initial peak of COVID-19 are at risk of long-term mental health problems even after the end of lockdown and removal of strong intervention measures during the initial peak [41]. In addition, COVID-19 resurgence is closely related to long-term mental health outcomes and a steeper longitudinal increase in symptom severity. Victims repeatedly exposed to similar disasters, such as earthquakes and tsunamis, as a result of these “repeated blows” are more susceptible to long-term distress [42,43]. Additionally, places hit by COVID-19 resurgences tend to be located adjacent to borders, which place them under persistent threat of imported cases and insecurity even when the domestic epidemic has been controlled [44,45]. Future studies are called for to further ascertain the mental health changing trend among people living in places severely affected by initial peak of COVID-19 or hit by COVID-19 resurgence over the long course of COVID-19.

Regarding work-related factors, self-perceived occupational exposure risk to COVID-19 and increases in work burden after resuming work were related with long-term mental health problems. People with high occupational exposure risk to COVID-19 were also psychologically vulnerable populations due to long-lasting fear, traumatic events and heavier workloads [9,46], which is consistent with previous studies of essential workers during epidemics [19,47,48]. In addition, approximately 40% of participants reported an increase in work burden after resuming work. High work load could lead to burnout, and enhance the risk of adverse mental health outcomes [49,50,51]. This points out that appropriate work allocation is necessary for prompting mental health during the COVID-19 pandemic and its recovery period.

We also found that participants obtaining psychological consultation were about three times more susceptible to long-term mental health symptoms, and over 70% of them reported distress at follow-up in the present study. After SARS a similar increase in seeking psychological consultation occurred [52], suggesting it was a potential warning signal for long-term distress following pandemics, and our study further indicated its potential in predicting a more pronounced temporal mental health deterioration over the course of COVID-19. Thus, those seeking psychological consultation deserve special attention for their mental health status during and after the initial peak of COVID-19 pandemic. Moreover, consistent with previous studies [53], family members of COVID-19 patients needed to be looked out for mental health symptoms. Specially, more understanding and support should be provided for families with deceased COVID-19 patients [54,55].

Protective factors that emerged were voluntarily wearing face masks when going out and reducing social gatherings. This may be due to the sense of security and altruism brought by self-protection measures during epidemics [56,57]. As positive attitudes towards and belief in effectiveness of self-protection measures can increase public compliance, the government is suggested to impart benefits of wearing face masks and keeping social distance to the public, which may bring about long-term merits [58,59]. However, the causal relationship between usage of self-protection measures and mental health should be interpreted with caution for those having fewer mental health problems may tend to comply with guidance on personal protective practices.

## 5. Strengths and Limitations

This study is among the earliest investigations tracking changes in mental health from the initial peak of COVID-19 to its aftermath. We come to our conclusions based on a nationwide and large cohort. The longitudinal within-subject design and repeated measurement on the same platform enabled us to explore the temporal changes with less chance of sampling and survey method bias. Additionally, we managed to longitudinally measure mental health symptoms across distinctive stages of COVID-19, and identified vulnerable populations that emerged during the initial peak and in its aftermath. Therefore, we believe our results can serve as a model for evolving mental health trends and offer some guidance for population- and phase-specific mental health management under the global enduring COVID-19 threat.

Our study has several limitations. These limitations include the observation time of only about five months, which might not be long enough to capture the complete psychological evolving trend during COVID-19. Moreover, evolving patterns and control levels of COVID-19 vary across countries, so application of our results to other countries should be made with caution. Future studies with longer follow-up time and from different countries are needed. Other limitations include the possible selection bias, low follow-up rate and failure to report reasons of drop-out. Although this study had extensive geographic coverage across China and a large sample size, selection bias still existed as all participants were paid members of a commercial website and were characteristically young and highly educated despite the fact that follow-up responders over-represented females, the elderly and the poorly educated, which warranted cautions when generalizing our results to larger populations. Finally, mental health symptoms were based on self-reported questionnaires instead of clinical diagnoses, so future studies employing more precise diagnostic methods are needed.

## 6. Conclusions

Our nationwide large-sample cohort study in China found susceptible populations during the initial COVID-19 peak suffered from enduring long-term mental health outcomes that still persisted in the aftermath of the initial peak. These populations included family members of COVID-19 patients, people with quarantine experiences, residents living in severely affected regions at the initial peak and people with occupational exposure risk to COVID-19. We further identified some new vulnerable populations that emerged after the initial peak, such as those living in places hit by COVID-19 resurgence, those seeking psychological consultation since COVID-19 and those experiencing increases in work burden after resuming work. Nonetheless, the significance of these results in the real world should be further validated in future researches. Our investigation reveals a severe and enduring mental health burden during the COVID-19 pandemic, suggesting essential long-term and regular mental health management throughout the whole course of COVID-19. This management will include making population- and phase-specific mental health intervention strategies during COVID-19.

## Figures and Tables

**Figure 1 ijerph-18-08790-f001:**
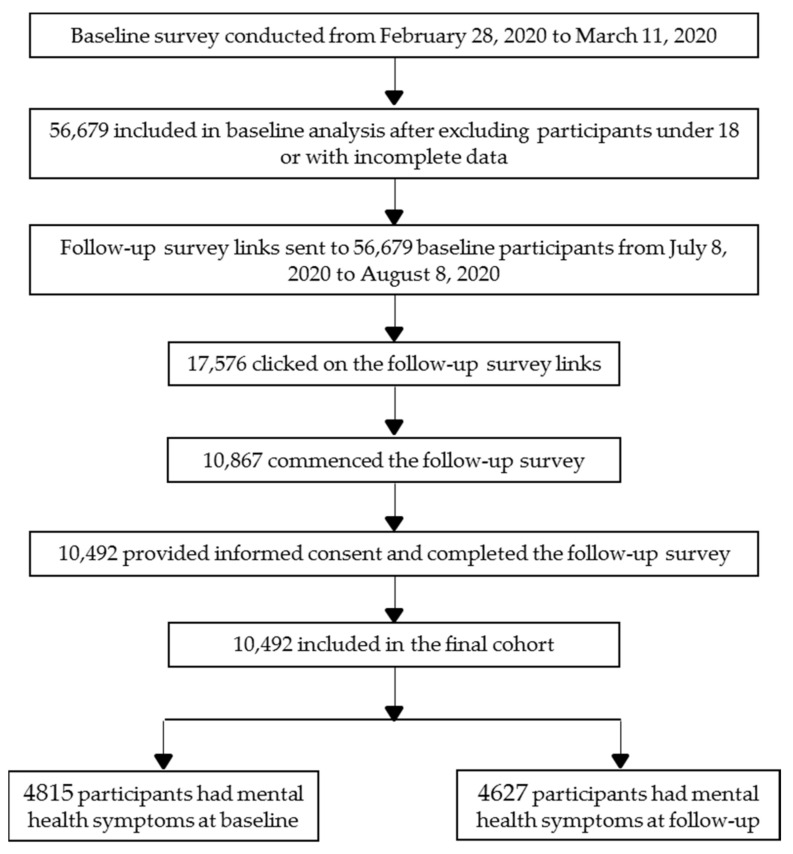
Flow chart of participants selection.

**Figure 2 ijerph-18-08790-f002:**
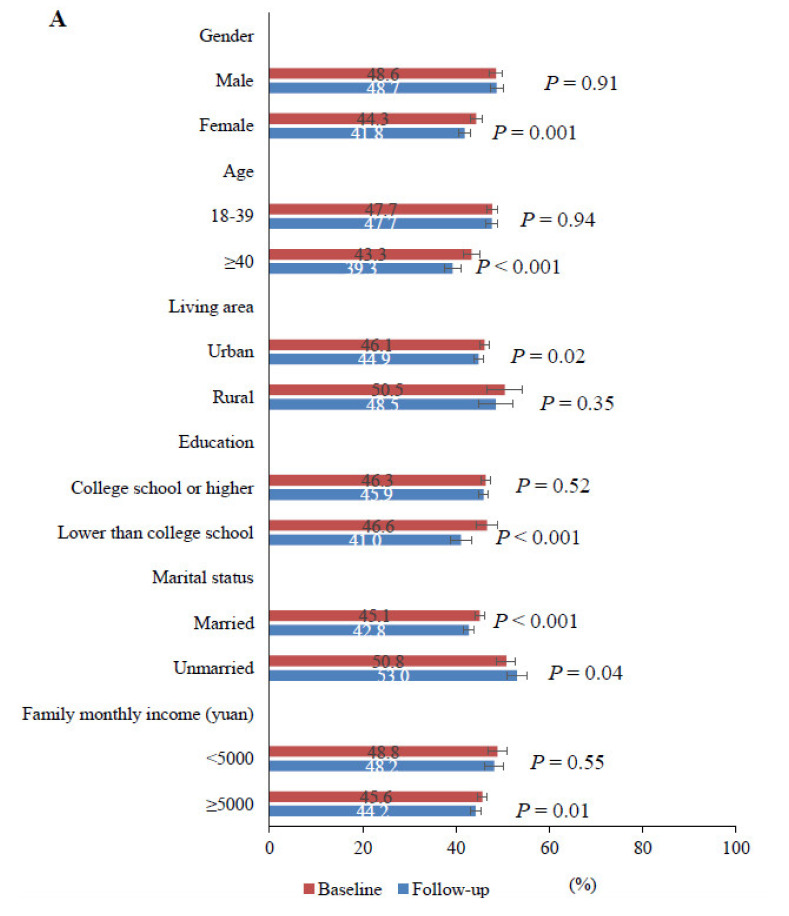
Changes in weighted prevalence of any mental health symptoms from baseline to follow-up survey. (**A**). Prevalence stratified by demographic-related factors. (**B**). Prevalence stratified by personal- and work-related epidemic factors. Any mental health symptoms is defined as having either depression, anxiety or insomnia. *p*-Values are from McNemar tests for statistically significant differences in prevalence of any mental health symptom in baseline vs. follow-up survey. The error bars indicate 95%CIs.

**Table 1 ijerph-18-08790-t001:** Baseline demographic characteristics of participants.

Factors	Total Number of Participants at Baseline (%)	Number of Baseline Participants Who Responded to Follow-Up Survey (%)	Number of Baseline Participants Who Did Not Respond to Follow-Up Survey (%)	*p*-Value ^a^	Weighted Number of Participants (%) ^b^
**Overall**	56,679 (100.0)	10,492 (100.0)	46,187 (100.0)	-	10,492 (100.0)
**Mean (SD) for age, years**	35.97 (8.22)	36.87 (8.21)	35.76 (8.21)	<0.001 ^c^	-
**Age, years**					
18–39	39,468 (69.6)	6815 (65.0)	32,653 (70.7)	<0.001	7306 (69.6)
≥40	17,211 (30.4)	3677 (35.0)	13,534 (29.3)		3186 (30.4)
**Gender**					
Male	27,149 (47.9)	4465 (42.6)	22,684 (49.1)	<0.001	5026 (47.9)
Female	29,530 (52.1)	6027 (57.4)	23,503 (50.9)		5466 (52.1)
**Living area**					
Urban	52,839 (93.2)	9769 (93.1)	43,070 (93.3)	0.60	9781 (93.2)
Rural	3840 (6.8)	723 (6.9)	3117 (6.7)		711 (6.8)
**Education**					
Lower than college school	9540 (16.8)	2101 (20.0)	7439 (16.1)	<0.001	1766 (16.8)
College school or higher	47,139 (83.2)	8391 (80.0)	38,748 (83.9)		8726 (83.2)
**Marital status**					
Married	43,763 (77.2)	8467 (80.7)	35,296 (76.4)	<0.001	8101 (77.2)
Unmarried	12,916 (22.8)	2025 (19.3)	10,891 (23.6)		2391 (22.8)
**Monthly income, yuan**					
0–4999	13,016 (23.0)	2444 (23.3)	10,572 (22.9)	0.37	2402 (22.9)
≥5000	43,663 (77.0)	8048 (76.7)	35,615 (77.1)		8090 (77.1)
**Geographical region**					
Eastern China	23,172 (40.9)	4283 (40.8)	18,889 (40.9)	0.01	4289 (40.9)
Northern China	10,227 (18.0)	1849 (17.6)	8378 (18.1)		1893 (18.0)
Northeastern China	3921 (6.9)	781 (7.4)	3140 (6.8)		726 (6.9)
Northwestern China	1348 (2.4)	219 (2.1)	1129 (2.4)		250 (2.4)
Central China	4803 (8.5)	873 (8.3)	3930 (8.5)		889 (8.5)
Southern China	10,028 (17.7)	1935 (18.4)	8093 (17.5)		1856 (17.7)
South west China	3156 (5.6)	547 (5.2)	2609 (5.6)		584 (5.6)
Missing values	24 (0.0)	5 (0.0)	19 (0.0)		4 (0.0)
**History of chronic diseases**					
Yes	3274 (5.8)	664 (6.3)	2610 (5.7)	0.007	606 (5.8)
Unknown/No	53,405 (94.2)	9828 (93.7)	43,577 (94.3)		9886 (94.2)
**History of psychiatric disorders**					
Yes	161 (0.3)	31 (0.3)	130 (0.3)	0.81	32 (0.3)
Unknown/No	56,518 (99.7)	10,461 (99.7)	46,057 (99.7)		10,460 (99.7)
**Family history of psychiatric disorders**					
Yes	396 (0.7)	64 (0.6)	332 (0.7)	0.23	64 (0.6)
Unknown/No	56,283 (99.3)	10,428 (99.4)	45,855 (99.3)		10,428 (99.4)

^a^*χ^2^* tests were used to compare proportion of participants who responded to the follow-up survey across different strata of demographic factors. ^b^ Weight was estimated based on gender (male or female), age (18–39 years or ≥40 years), living area (urban or rural), educational level (lower than college school or college school or higher), marital status (married or unmarried), geographical regions and history of chronic diseases (yes or unknown/no) using the baseline sample. ^c^
*p*-Value from *t* test.

**Table 2 ijerph-18-08790-t002:** Prevalence and scores of depression, anxiety, insomnia and any mental health symptoms at baseline and follow-up survey.

Mental Health Symptoms	Number of Participants with Mental Health Symptoms (%, 95%CI)					Median Scores (IQR)		
	Baseline (N = 10,492)		Follow-Up (N = 10,492)		*p*-Value ^b^	Baseline	Follow-Up	*p*-Value ^c^
	Unweighted	Weighted ^a^	Unweighted	Weighted ^a^				
Depression	3071 (29.3, 28.4–30.2)	3151 (30.0, 29.2–30.9)	3421 (32.6, 31.7–33.5)	3528 (33.6, 32.7–34.5)	<0.001	0.0 (0.0–6.0)	0.0 (0.0–8.0)	<0.001
Anxiety	3654 (34.8, 33.9–35.7)	3693 (35.2, 34.3–36.1)	3320 (31.6, 30.8–32.5)	3415 (32.5, 31.7–33.5)	<0.001	1.0 (0.0–7.0)	0.0 (0.0–7.0)	<0.001
Insomnia	3066 (29.2, 28.4–30.1)	3127 (29.8, 28.9–30.7)	3610 (34.4, 33.5–35.3)	3701 (35.3, 34.4–36.2)	<0.001	4.0 (1.0–8.0)	4.0 (1.0–9.0)	<0.001
Any mental health symptoms	4815 (45.9, 44.9–46.9)	4865 (46.4, 45.4–47.3)	4627 (44.1, 43.1–45.1)	4733 (45.1, 44.2–46.1)	0.01	-	-	-

^a^ Weight was estimated based on gender (male or female), age (18–39 years or ≥40 years), living area (urban or rural), educational level (lower than college school or college school or higher), marital status (married or unmarried), geographical regions and history of chronic diseases (yes or unknown/no) using the baseline sample. Depression is defined as having a Patient Health Questionnaire-9 score ≥ 5. Anxiety is defined as having a Generalized Anxiety Disorder-7 score ≥ 5. Insomnia is defined as having an Insomnia Severity Index score ≥ 8. Any mental health symptoms is defined as having either depression, anxiety or insomnia. ^b^
*p*-Values from McNemar *χ^2^* tests comparing weighted prevalence at baseline and follow-up survey. ^c^
*p*-Values from paired-samples Wilcoxon tests comparing scores at baseline and follow-up survey.

**Table 3 ijerph-18-08790-t003:** Proportions of new onset and persistent depression, anxiety, insomnia and any mental health symptoms from baseline to follow-up survey in the cohort.

Mental Health Symptoms	Proportion of New Onset Symptoms among Baseline Negative Participants		Proportion of Persistent Symptoms among Baseline Positive Participants	
	Unweighted n/N (%)	Weighted n/N (%) ^a^	Unweighted n/N (%)	Weighted n/N (%) ^a^
Depression	1445/7421 (19.5)	1475/7341(20.1)	1976/3071 (64.3)	2052/3151 (65.1)
Anxiety	1220/6838 (17.8)	1256/6799 (18.5)	2100/3654 (57.5)	2159/3693 (58.5)
Insomnia	1528/7426 (20.6)	1562/7365 (21.2)	2082/3066 (67.9)	2139/3127 (68.4)
Any mental health symptoms	1346/5677 (23.7)	1374/5627 (24.4)	3281/4815 (68.1)	3359/4865 (69.0)

^a^ Weight was estimated based on gender (male or female), age (18–39 years or ≥40 years), living area (urban or rural), educational level (lower than college school or college school or higher), marital status (married or unmarried), geographical regions and history of chronic diseases (yes or unknown/no) using the baseline sample.

**Table 4 ijerph-18-08790-t004:** Associated factors of positive of long-term depression, anxiety, insomnia and any mental health symptoms during the course of COVID-19 in the multivariable generalized estimating equation models ^a^.

Factors	Depression		Anxiety		Insomnia		Any Mental Health Symptoms	
	OR (95%CI)	*p*-Value	OR (95%CI)	*p*-Value	OR (95%CI)	*p*-Value	OR (95%CI)	*p*-Value
Gender: male (vs. female)	1.19 (1.11–1.28)	<0.001	1.04 (0.97–1.12)	0.30	1.18 (1.10–1.27)	<0.001	1.08 (1.01–1.16)	0.03
Age(years)	0.99 (0.98–0.99)	<0.001	0.99 (0.99–1.00)	<0.001	1.00 (1.00–1.01)	0.62	1.00 (0.99–1.00)	0.08
Marital status: married (vs. unmarried)	0.85 (0.77–0.94)	0.001	0.95 (0.86–1.04)	0.25	0.78 (0.71–0.86)	<0.001	0.83 (0.75–0.91)	<0.001
Family members of COVID-19 patients: yes (vs. no)	1.79 (1.38–2.33)	<0.001	1.75 (1.37–2.22)	<0.001	1.86 (1.43–2.42)	<0.001	2.10 (1.61–2.74)	<0.001
Experiences of quarantine: yes (vs. no)	1.35 (1.25–1.46)	<0.001	1.34 (1.24–1.44)	<0.001	1.29 (1.20–1.40)	<0.001	1.31 (1.22–1.41)	<0.001
Living in province most severely affected by initial peak: yes (vs. no)	1.28 (1.10–1.49)	0.002	1.32 (1.14–1.53)	<0.001	1.19 (1.02–1.39)	0.03	1.21 (1.04–1.41)	0.01
Self-perceived occupational exposure risk to COVID-19: yes (vs. no)	1.33 (1.20–1.48)	<0.001	1.39 (1.26–1.54)	<0.001	1.18 (1.06–1.31)	0.002	1.26 (1.14–1.40)	<0.001
Living in places with COVID-19 resurgences: yes (vs. no)	1.36 (1.24–1.49)	<0.001	1.30 (1.18–1.42)	<0.001	1.37 (1.25–1.50)	<0.001	1.38 (1.26–1.50)	<0.001
Increases in work burden after resuming work: yes (vs. no)	1.79 (1.66–1.92)	<0.001	1.76 (1.64–1.90)	<0.001	1.78 (1.66–1.92)	<0.001	1.77 (1.65–1.90)	<0.001
Wearing face masks voluntarily when going out: yes (vs. no)	0.70 (0.58–0.85)	<0.001	0.71 (0.59–0.87)	<0.001	0.75 (0.62–0.90)	0.003	0.74 (0.61–0.90)	0.003
Reducing social gatherings voluntarily: yes (vs. no)	0.84 (0.74–0.95)	0.004	0.92 (0.82–1.03)	0.16	0.83 (0.74–0.94)	0.003	0.93 (0.83–1.04)	0.22
Seeking psychological consultation since COVID-19: yes (vs. no)	2.81 (2.55–3.10)	<0.001	2.78 (2.52–3.06)	<0.001	2.36 (2.14–2.60)	<0.001	2.57 (2.33–2.85)	<0.001

^a^ Values were from multivariable generalized estimating equation models with binomial distribution, the survey order as repeated effect and within-subject effect, and participant-specific ID code as covariate factor, using an autoregressive correlation structure. All factors considered that showed significance in the univariable generalized estimating equation models were entered into the multivariable models.

**Table 5 ijerph-18-08790-t005:** Associated factors of scores of depression, anxiety and insomnia symptoms during the course of COVID-19 in the multivariable generalized linear mixed models ^a^.

Factors	Depression		Anxiety		Insomnia	
	b (SE)	*p*-Value	b (SE)	*p*-Value	b (SE)	*p*-Value
Gender: male (vs. female)	0.37 (0.08)	<0.001	0.06 (0.07)	0.39	0.34 (0.09)	<0.001
Age (years)	−0.02 (0.01)	<0.001	−0.01 (0.00)	0.007	0.01 (0.01)	0.15
Marital status: married (vs unmarried)	−0.38 (0.11)	<0.001	−0.02 (0.09)	0.82	−0.70 (0.12)	<0.001
Family members of COVID-19 patients: yes (vs. no)	1.48 (0.30)	<0.001	1.44 (0.25)	<0.001	1.99 (0.31)	<0.001
Experiences of quarantine: yes (vs. no)	0.74 (0.08)	<0.001	0.64 (0.07)	<0.001	0.58 (0.09)	<0.001
Living in province most severely affected by initial peak: yes (vs. no)	0.81 (0.18)	<0.001	1.04 (0.16)	<0.001	0.71 (0.20)	<0.001
Self-perceived occupational exposure risk to COVID-19: yes (vs. no)	0.87 (0.12)	<0.001	0.70 (0.10)	<0.001	0.38 (0.13)	0.003
Living in places with COVID-19 resurgences: yes (vs. no)	0.69 (0.10)	<0.001	0.45 (0.09)	<0.001	0.83 (0.11)	<0.001
Increases in work burden after resuming work: yes (vs. no)	1.07 (0.08)	<0.001	1.02 (0.07)	<0.001	1.27 (0.09)	<0.001
Wearing face masks voluntarily when going out: yes (vs. no)	−0.93 (0.23)	<0.001	−0.56 (0.20)	0.004	−0.72 (0.24)	0.003
Reducing social gatherings voluntarily: yes (vs. no)	−0.51 (0.14)	<0.001	−0.15 (0.12)	0.21	−0.22 (0.15)	0.14
Seeking psychological consultation since COVID-19: yes (vs. no)	2.72 (0.12)	<0.001	2.03 (0.11)	<0.001	1.82 (0.13)	<0.001
Educational level × time: college school or higher (vs. lower than college school)	0.02 (0.01)	0.03	0.03 (0.01)	0.003	0.05 (0.01)	0.001
Living in places with COVID-19 resurgences × time: yes (vs. no)	0.03 (0.01)	0.003	0.04 (0.01)	0.002	0.03 (0.02)	0.10
Increases in work burden after resuming work × time: yes (vs. no)	0.04 (0.01)	<0.001	0.02 (0.01)	0.04	0.09 (0.01)	<0.001
Wearing face masks voluntarily when going out × time: yes (vs. no)	−0.07 (0.03)	0.005	−0.07 (0.03)	0.02	−0.05 (0.04)	0.17
Reducing social gatherings voluntarily × time: yes (vs. no)	−0.02 (0.01)	0.15	−0.05 (0.02)	0.003	−0.08 (0.02)	<0.001
Seeking psychological consultation since COVID-19 × time: yes (vs. no)	0.21 (0.02)	<0.001	0.19 (0.02)	<0.001	0.21 (0.02)	<0.001

^a^ Values were from multivariable generalized linear mixed models with random within-subject intercepts and autoregressive covariance structures. The multivariable models included fixed effects for time (coded as 0 at baseline, and 5 at follow-up), as well as all factors and their interaction terms with time that showed significance in their individual univariable generalized linear mixed models.

## Data Availability

The corresponding authors have full access to all the data in the study and take responsibility for the integrity of the data and the accuracy of the data analysis.

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
