# Peer review of "Long-Term Impact of COVID-19 on Mental Health among the General Public: A Nationwide Longitudinal Study in China"

_ijerph, 2021, doi:10.3390/ijerph18168790_

Round 1
Reviewer 1 Report
Dear authors,
I believe that "Long-term impact of COVID-19 on mental health: a naionwide longitudinal study in China, is a quality article.
The study is very interesting , the research design is appropiate and the results are clearly presented. However, the introduction can be improved.
My proposed change would be as follows:
Introduction
The introduction don´t provide sufficient background. It would be interesting to add the following topics:
- It is necessary to study mental health from a gender perspective. Studies have shown that factors and prevalence is different between men and women.
- Write in the introduction the findings of research from other countries on COVID and mental health.
- previous studies after SARS indicate potential long-term mental health repercussions after pandemics : and gender differences ? and which variables explained mental health ? If the effect of some variables is analyzed, it is necessary to show antecedents in the introduction.
Objectives and Hypotheses
It would be appropriate to write explicitly the objectives and hypotheses.
Variables
Age. Why only two groups? It is a very large range 18-39, and over 40 years old.
Reviewer 2 Report
The data are generally well analysed and presented. However, the authors should provide more information on the nature of their sample (users of Joybuy) and how this sample would differ from the general population - and hence the implications for interpreting the results. They could also provide information on any available mental health statistics prior to COVID.
Reviewer 3 Report
This is an interesting paper on an important topic, relevant to readers across the world due to the ongoing Covid-19 pandemic. The authors conducted a longitudinal study about the effects of the pandemic on mental health, in particular depression, anxiety and insomnia. They have acquired an impressive sample of participants, making it more likely that the findings within the manuscript can be generalized to a larger population.
I have a few questions/comments, which I hope make the article stronger:
- From the title and also the introduction, I initially got the impression that the study would be about individuals who had been infected with Covid-19 and their long-lasting mental health issues. Upon reading the 2nd half of the introduction, it became clear that the research is focused more on the impact of the measurements/restrictions and pandemic in general. I think the authors should make this more clear from the start and in the title.
- The authors refer to the fact that the precise recruitment method was described elsewhere, but I feel it also needs more elaboration in the current paper. Why for example choose this platform as a recruitment place? For non-Chinese readers, it is not clear what kind of website JoyBuy is. Also, registered members were sampled - are these different from the non-members in any way? For example, bias could enter the sample if it costs money to be a member. More information is needed here, both -briefly- about the platform and its members and the general recruitment strategy. E.g. inclusion or exclusion criteria are not mentioned.
- Was participation voluntary or did participants receive anything in return?
- It is unclear to me why so many of the variables were dichotimized, and whether these were measured like this, or constructed like this after data collection was finished. It seems that variance is lost if age is measured with 2 answer options rather than a continuous measure. Could the authors elaborate on their approach?
- Were the surveys at Time1 (baseline) and Time 2 exactly the same? Or were there some changes?
- I think the authors could have chosen a more strict CI e.g. 99% CI (or significance level .01)
- The authors should reflect on the ecological validity and significance of their findings - due to their large sample, even small changes and differences were highly significant, but it remains a question whether these are 'ecologically' significant as well. Perhaps providing the effect sizes of your analyses would help in this regard as well.
Reviewer 4 Report
Comments:
Thank you for the opportunity to review this interesting manuscript. I have a few minor comments for the authors to consider:
- Abstract:
- Lines 36-40: Please report the adjusted (or crude) ORs and the 95% CIs for the significant factors.
- Introduction:
- The authors indicated that there are many factors (line 72) that are associated with immediate mental health symptoms, which may induce sustained mental health changes. As the authors stated the aim of their study in the last sentence of this section, it would be good to briefly mention the scope of the factors. The authors could just add the examples, e.g. environmental, societal, economical, etc in a bracket to set the scope of this study.
- Methods:
- Section 2.1: Please justify the choice of using the Joybuy website as this study’s sampling frame. The study is likely to miss people who are not patronage of Joybuy, and perhaps people of certain socioeconomic status. It would be good to also provide some data about the representativeness of the Joybuy users against that of the whole population in China.
- Line 112: The authors cited reference 4 about the measures. It would be good to report the validity and reliability of the questionnaires, or the specific types of validity and reliability the authors investigated (E.g. back translation for the Chinese PHQ-9, etc).
- Line 154: Please state the level of significance – was it 0.05 or 0.10 or 0.20?
- Results:
- Section 3.4 (Table 4) and Section 3.5 (Table 5): The authors may choose to keep only 1 as the information are redundant. The authors may just keep Section 3.4 and Table 4 if the clinicians are the target audience.
- Strengths and limitations:
- In the second paragraph, the first sentence could be revised to just “Our study has several limitations”. The authors mentioned the low response rate. It would be good to also add another limitation in relation to this, i.e. this study could not report the reasons of lost to follow-up – could it be that they died, or they had severe (mental) illnesses that prevented them from participating in the follow-up data. The authors could also comment on the validity and reliability of the measurements if they were relatively low.
Round 2
Reviewer 1 Report
It is OK
Reviewer 2 Report
The manuscript has been markedly improved.